# Porous Secularity: Religious Modernity and the Vertical Religious Diversity in Cold War South Korea

## Kyuhoon Cho

Department of Gender, Religion and Critical Studies, University of Regina, Regina, SK S4S 0A2, Canada; kyuhoon.cho@uregina.ca

**Abstract:** Beyond the once dominant secularization thesis that anticipated the decline of religion in the modern era, the academic study of religion has in recent decades revisited secular as one of the factors that shape religion and religions in the globalized world. Against this theoretical backdrop, in this article, I use the case of South Korea to explore how secular and religion interact in contemporary global society. It focuses on describing the postcolonial reformulation of secularity and the corresponding transformation of religious diversity in Cold War South Korea. The Japanese colonial secularism rigidly banning the public and political engagement of religion was replaced by the flexible secular-religious divide after liberation of 1945. The porous mode of secularity extensively admitted religious entities to affect processes of postcolonial nation-building. Religious values, interests, and resources have been applied in motivating, pushing, and justifying South Koreans to devote themselves to developing the national community as a whole. Such a form of secularity became a critical condition that caused South Korea's religious landscape to be reorganized in a vertical and unequal way. On the one hand, Buddhist and Christian populations grew remarkably in the liberated field of religion, while freedom of religion was recognized as a key ideological principle of the anticommunist country. On the other hand, folk beliefs and minority religious groups were often considered "superstitions", "pseudo religions", "heretics", or even "evil religions". With the pliable secularity at work, religious diversity was reconfigured hierarchically in the postcolonial society.

**Keywords:** porous secularity; modernities; religious diversity; religious sphere; South Korea

## 1. Introduction

The modern era is characterized by a shift from a religious or transcendent order to a secular, natural, or immanent frame (Taylor 2007, pp. 539–44). This general process, through which the influence of religion declines, is often called secularization. From a global perspective, this form of secularization experienced by the West can be said to be one of the various processes that have led to the secular-religious divide (Burchardt et al. 2015, pp. 1–7). Beyond the once dominant secularization thesis that anticipated the modern decline of religion, the academic study of religion has in recent decades revisited secular as one of the critical factors that shape the location of religion and religions in the modern world.[1]

Secularity, i.e., a mode of decoupling social spheres from religious norms and institutions, may take shape responding to societal problems such as securing individual freedom, balancing religious heterogeneity, achieving national integration, and maintaining the autonomous operation of differentiated social domains. How does secularity engage in religion and religions in the globalized East? How does globalization relate to the interplay between religion and secular in non-Western societies? This article is intended to answer these questions through the case of South Korea.

The local manifestations of religion in contemporary times are closely related to, or even influenced by, the emergence of the global system of nation-states, many of which are considered secular. In the modern East Asian sphere, the foremost problem for most

Koreans to accept the secular-religious divide was to protect, restore, and/or develop their national community in crisis in the globalized world. As East Asian countries were incorporated into the global political order, the Western idea of secularity was disseminated to Korea, which, in turn, triggered a rise and use of the term *chonggyo* 宗教[2], the pan-East Asian translation of the English term 'religion'. The modern notion of 'religion' that refers to a distinctive field of communication first appeared in Korea in the historical course through which the national society became part of the globalized modern world. In the late 19th and 20th centuries, what was called 'religion' was widely expected to assume a distinctive identity, status, or role that was different from those of other secular sectors, groups, or institutions, all of which together constituted modern Korea.

The re-entry of the East Asian region into the global Cold War order triggered the reconstruction of secularity, which was, in turn, greatly attributed to the formation of religious modernity in the southern part of the Korean Peninsula. In what form, then, was the secular-religious divide formulated in the processes of South Koreans' nation-building? This article is aimed at examining how secularity was distinctively connected and interacted with the religious field in Cold War South Korea. In particular, it focuses on describing the postcolonial reformulation of secularity and the corresponding discursive and organizational transformation of religion and religions (Dreßler 2019, pp. 3–6) in South Korean society from 1945 to the 1980s/90s. To capture the comparative characteristics of South Korean secularity, this paper heuristically invokes the multiple secularities framework that relativizes the Western processes of secularization (Kleine and Wohlrab-Sahr 2016; Wohlrab-Sahr and Burchardt 2012). I attempt to distinguish the case of South Korea in terms of cultural and historical imprints, Cold War geopolitics, critical regional and national junctures, and national guiding ideas.

## 2. The Reconfiguration of Secularity in Postcolonial South Korea

### 2.1. Secular Modernization and the Contribution of Religion in the Global Cold War[3]

The Japanese annexation of Korea (1910–1945) was aimed at perpetuating imperial Japan with the 'divine emperor' at its peak. Under the guiding idea of State Shintō, which was officially 'secular', the Japanese colonial government implemented laws and policies that strictly divided public institutions and social fields, such as public schools, hospitals, and science and technology, from religious organizations and customs. In 1945, in the end, the Japanese occupation of Korea came to an end.

What changes, then, did liberation bring about in the secular-religious distinction in Korea? What were the cultural foundations, historical memories, and global forces that reshaped South Korean secularity in the latter half of the 20th century? Roughly speaking, the rigid secularism of the Japanese colonial government that had strictly restricted Koreans' religious activities to the private sector, after the liberation, turned into a rather flexible or fluid mode of secularity, one in which in the southern part of the Korean Peninsula, religious individuals and groups were extensively allowed or even invited to participate in civic movements, social development projects, party politics, and/or state governance (Noh 2010, pp. 18–31). Such a postcolonial transformation of secularity is attributed to a series of socio-political changes and underlying cultural elements, including liberation, global Cold War politics, South Korea's relationship with the Western liberal bloc, the path-dependent remnants of the Japanese occupation, and the traditional religio-cultural heritages.

With the demise of imperial Japan, a new regional order unfolded in East Asian region, which resulted in not only the tragic division of Korea into two political entities, i.e., North Korea and South Korea, based on opposing ideological lines, but also the subsequent establishment of a single separate state in the southern part of the Korean Peninsula. The emancipation of Korea from Japanese colonialism did not restore Confucianism—the governing ideology of the Chosŏn Dynasty (1392–1910)—as the state religion. The rise of Cold War geopolitics and the augmented affinities between Korean nationalism and anticolonial religious aspirations, instead, were crucial in making this new independent coun-



try, especially its laws, regulations, and institutions, favorable for religious organizations in South Korea.

The American troops entered the southern part of Korea in 1945.[4] Japanese secularism, while placing State Shintō at the heart of the colonial strategy of ruling Koreans, had extensively suppressed Korean religious groups from participating in political and even public activities. In contrast to that, the United States Army Military Government in Korea (USAMGIK, 1945–1948) not only officially stipulated "there should be no state religion" like State Shintō, but also further widely invited religious individuals and societies to assist the USAMGIK in a bid to build the young country and protect it from the northern communists (Kang 2013, pp. 28–32). Those laws and policies, which had widely allowed the public engagement of religions while formalizing the separation of state and religion during the three years of US military rule, were, without any fundamental change, continued by the subsequent governments of the Republic of Korea.

Establishing a unified nation-state on the Korean Peninsula became the core task in two Koreas—a primary thrust that accelerated the secular development of Korean societies in the Cold War world order. "The reunification and regeneration of the divided fatherland"[5] was the central guiding idea of South Korean society, which was over time solidified by a series of political upheavals, economic challenges, and ideological or cultural constructs such as the Korean War, developmental dictatorships, anticommunism, ethnic nationalism, and Confucian familism. It does not necessarily mean that, except for the mission of reunifying the fatherland and making it prosperous, there were no other critical problems that could impact the location of religion and religions in this postcolonial society. Rather, most religious issues were, after all, observed and evaluated from that anticommunist-nationalist-modernist perspective of whether religion could be helpful or a hindrance to the survival and development of South Korea in the globalized modern world. As the nationalist effervescence to create a single robust state prevailed in the Korean Peninsula, there rarely existed any political space available for different religious groups obstructing the governance of this fledgling nation or for inter-religious clashes or religion-related problems causing extensive nationwide splits.

Therefore, the inter-Korean tensions became an underlying problem in promoting the worldly functional development of postcolonial society on the one hand and encouraging the public and even political participation of religions in South Korea on the other. With the national agenda of anticommunism, the First Republic (1948–1960) whose president was Syngman Rhee (r. 1948–1960), a Methodist elder who was receiving widespread support from the Protestant circles, had no hesitation in cooperating with, or even giving preference to, religious organizations, Protestant and Catholic groups in particular. To win the 'regime competition' against the northern communists, religious groups were extensively mobilized to take part in civil society and the public arena at large, contributing to the advancement of educational, medical, mass media, and welfare institutions, as well as women's leadership. Religious organizations, manpower, and resources were largely requested and utilized to build local schools, universities (Cho 2006, pp. 100–4), and modern hospitals, support critical newspapers and broadcasters monitoring the activities of state institutions, and maintain orphanages for war orphans (Clark 2007, pp. 174–75). Nevertheless, the social engagement of religions did not mean that religions should interfere in the inherent operation of differing social sectors and institutions, but rather that they should support or sometimes lead the functional enhancement of those social domains according to their respective logics and roles for the national society (Kang 2013, pp. 45–56). In any case, again, the public participation of religions was not a throwback to the past where a dominant religion such as Confucianism reigned over everything (Baker 2013, p. 185), but, for many South Koreans, it meant acquiring meaning and direction to get empowered and move forward in a newly urbanized capitalist society.

The war and subsequent state-led economic development made South Korea's social and cultural landscape very urban and secular. To reunify the divided nation according to their respective ideologies, North and South Koreans went to war from 1950 to 1953—the

first hot battle of the Cold War. The Korean War desacralized many Koreans as the urgency of modern warfare made a mockery of conventional Confucian manners, as well as religiously rooted norms, values, and customs. With the end of the Korean War, a great degree of community sentiment that the establishments, rural communities, and traditional cultures had cherished was replaced by a strong sense of hostility based on ideological differences and profound individual and collective wounds. The Korean War turned the vibrant expectation of liberation of 1945 into a nightmare. It resulted in the casualties of approximately 650,000 soldiers and 2.5 million civilians in the North and South, including numerous massacres of civilians. The quasi-racist ideology of anticommunism justified the mass killings. Syngman Rhee, the ROK president at the time, dehumanized those citizens who were considered communists as "the enemies of human society". South Korean police and military often killed "suspected communists" without distinguishing the innocent, whether children, women, or the elderly, from the enemy (Kim 2004, pp. 536–39).[6]

The postwar industrialization was a sort of disenchantment process of the nation-state that changed its cultural landscape from a spiritual, pastoral, and communal milieu to a worldly, materialist, and individualist one in the latter half of the 20th century. The Park Chung-hee regime (r. 1961–1979) started its signature initiative, *Saemaul Undong* (새마을 운동, New Village Movement), in the 1970s, a state-led modernization project that was often publicized to "develop rural villages and defeat superstitions". It sparked a dramatic social and environmental transformation in this underdeveloped East Asian country within three decades. Most South Koreans who had lived in rural village communities until the 1960s came to live in a new urban environment in the 1990s.[7] When anthropologist Laurel Kendal interviewed a female Korean shaman in the late 1970s, she deplorably testified to the spiritual effects of this socio-economic transformation as follows:

> "Nowadays the mountains just don't have the same power to give us inspiration [*myŏnggi*] anymore. … Nowadays, we have to make a thousand prostrations to the gods and ask them, 'Should this particular family do such and such?' In the past, it just came to us without even asking". (Kendall 2009, p. xvii)

While the authoritarian state was suppressing shamanic rituals performed in urban outskirts and rural areas as 'superstitions' that were the enemy of *Chokuk kŭndaehwa* (조국근대화, modernization of the fatherland), industrialization evaporated the shamanistic sacredness that Koreans had felt daily from the natural environment, especially its overwhelming energy. In the rapid reformulation of the socio-economic landscape, centuries-old Shamanic folk belief, the cultural and religious grounds of rural communities, markedly disappeared, and the disenchantment of the life-world, in turn, was made to a great extent. Leaving behind the spirit-filled rural culture for good, contemporary South Korea has come to be characterized by a highly urbanized culture, as often represented by K-Pop or Seoul's Gangnam district, a symbolic centre of South Korean capitalism.[8]

### 2.2. Porous Secularity for the Sake of Nation-Building

How were religious elements connected to the secular transformation of the Republic of Korea after liberation? Although modern instrumental rationality increasingly replaced the conventional plausibility structure provided by religious traditions such as folk beliefs and Confucianism, the religious impacts did not completely disappear in the rapid industrialization and urbanization processes. That said, the contribution of religions to the social changes in postcolonial South Korea was not limited to the institutional and organizational levels; a sort of 'moral order' (Taylor 2004, pp. 3–30) based on religious norms or values survived, providing South Koreans with a fundamental motivation to carry out a transition from a traditional rural community to a developed industrial economy. For example, Confucian ethic and imaginaries were highly influential in driving the sweeping social transformation, often called *Kŭndaehwa* (근대화, modernization), which was largely planned and led by the developmental state.

Certain 'collective effervescence' or 'consciousness of obligation' that seemed rooted in the Confucian moral code greatly accelerated South Koreans to devote themselves to ma-

terialistic or quantitative goals, contributing to economic growth and the further functional advancement of social institutions. As the Protestant ethic played a role in the rise of modern capitalism in Western Europe (Weber [1930] 2005, pp. 1–13), Confucian aspirations and ideals imposed on South Koreans the moral imperative that held the key to the emergence of Korean modernity. Such longing for these worldly objectives could be articulated in such secular formations as nationalism, anticommunism, developmentalism, and communalism. The core ethical idea underlying these secular ideologies is that individuals should observe a series of behavior principles such as 'diligence', 'frugality', 'self-restraint', 'sacrifice', 'loyalty', 'discipline', and 'self-cultivation', in a bid to boost profits and uphold honor for their affiliated groups such as family, local community, company, political faction, and most importantly, the fatherland. Namely, the tremendous passion for institutional rationality, secular political economy, and quantitative industrial growth in Cold War South Korea was largely rooted in the moral regime for which one may name 'Confucian values' or 'Confucian familism' (Chang 2010, pp. 1–13). Behind the guidelines to behave properly as a factory worker, salesperson, physician, politician, stockbroker, artist, entrepreneur, professional, or democratic citizen, there was a Confucian ethical standard that called for honoring one's family, company, or country and saving face.

In many perspectives, Japanese colonial rule, which Koreans once resisted and still denounce, preceded South Korea's modernity in applying Confucian familial images to govern the country. The Japanese colonial system, which placed the emperor, the 'true parent', above everything, shows significant similarities to the way the dictatorial state developed South Korean society, especially the way it used Confucian norms and religious organizations as a means of social engineering. It is true that, unlike imperial Japan, the Republic of Korea did not create a state religion and generally tolerated or even encouraged public participation of different religions; however, it is the same as imperial Japan in that the state, the authoritarian leaders in particular, as a kind of 'moral teacher' for the nation stood above any religions. The growth of the South Korean economy is often encapsulated by the so-called 'state-led capitalism', which greatly utilized the nation's collective culture based on Confucian familism. Authoritarian political leaders, e.g., Park Chunghee, Chun Doo-hwan (r. 1980–1987), and Roh Tae-woo (r. 1987–1994), all of whom were former Army generals, upheld 'developmental dictatorship', exquisitely manipulating the Confucian views of human relations to justify their undemocratic ruling.[9] Especially, they defined South Korean people as an extended family. No matter what religious organization or tradition they subscribed to, the ROK citizens were given the moral obligation to be loyal to the state's order for the glory of the familial society. Although close exchanges with Western advanced societies introduced them to such modern principles as 'freedom', 'human rights', 'equality', and 'liberal democracy', traditional religious values and aspirations still prevailed.

Such entangled modernity as it was constructed in postcolonial South Korea primarily referred back to the patriarchal authoritarian system of Japanese colonial rule. A direct developmental model for the fledgling nation pursued by the aforementioned military dictators was *Manchukuo* (滿洲國, 1932–1945), a puppet state of imperial Japan in northeast China and Inner Mongolia.[10] As the Japanese colonialists did in *Manchukuo* some decades earlier, the authoritarian government focused its institutional support on a series of selected companies that provided bribes or closely met its secrecy demands. The so-called *chaebol* (財閥) such as Hyundai or Samsung,[11] which had begun business as a rice retailer or small trading company during colonial times, grew rapidly, receiving preferential treatment from the ROK government. The foreign exchange crisis of 1997, from which South Koreans greatly suffered, has been blamed on corruption, cozy relations between politics and business, and/or cronyism, all of which were often credited to Confucian familism. The South Korean conglomerates even attempted to maintain the image of a 'true family' which has been a culturally well-accepted sales and management tactic.

In short, the Cold War geopolitics of globalized East Asia greatly contributed to the reformulation of secularity in South Korea, making it pliable for religions at both the cul-

tural and institutional levels. On the one hand, unlike the Chosŏn Dynasty and imperial Japan that had once ruled Korea, the Republic of Korea maintained no state religion that all citizens were obliged to follow. In the process of the functional and institutional development of the postcolonial society, the religious milieu of local communities, which was full of elements of folk beliefs, was quickly replaced by the urban, earthly, and material culture. On the other hand, religions played a role in the secular development of different social institutions and fields in that nation-state. In the name of religious freedom, the civil engagement of religious organizations is largely guaranteed by the ROK's legal and political systems. A prevalent sense of Confucian morality was a major cultural resource that led to the secular functional transformation of South Korean society. Confucian moral codes and aspirations had an influence on both positive and negative aspects of the formation of South Korean modernity. In the next section, I examine the reformulation of the religious topography in Cold War South Korea, which was greatly stimulated by the postcolonial transformation of secularity.

### 3. Vertical Division of Religious Diversity in an Entangled East Asian Modernity

*3.1. Fluid Secularity for Christian and Buddhist Traditions*

The formation of porous secularity was accompanied by a religious change in the postcolonial society. The 1945 liberation, in the end, enabled Korean religious communities to freely communicate with both domestic and foreign partners that were believed to share the same religious roots, networks, and/or theologies. The geopolitical importance of South Korea on the front lines of the Cold War made the religious sector of this divided country highly receptive to the institutional, material, and cultural impacts of the Western advanced societies, especially the USA. The enactment of laws regarding religion would be one of the representative examples. Once taking control of the southern part of the Korean Peninsula, the American military government declared to abolish discriminatory acts and regulations on faith, which had been created during the Japanese colonial rule. On 9 September 1945, the USAMGIK proclaimed Ordinance 11, whose Section 2 stated as follows: "All other laws, decrees or orders having the force of law are hereby repealed, the Judicial or Administrative enforcement of which would cause discriminations on grounds of … creed or political opinions". The emergence of amicable relations between religion and state provided a crucial institutional basis for fostering the growth and innovation of the religious sector, especially religious organizations, populations, and discourses.

Under the overwhelming socio-cultural influence of the USA that led the so-called 'First World' during the Cold War, a flexible mode of differentiation between religious and other social spheres, including the political one, was established in South Korea, which, in turn, created favorable conditions for many religious individuals and groups to engage in a wide panoply of public or secular activities. While in China and Japan, 'religion' was extensively signified as contradicting their respective projects of secular nation-building, in South Korea, there, by and large, appeared discourses and measures in the political, legal, educational, and media spheres that affirmed and even called for the contribution of religion to the nation's advancement and the reunification of two Koreas. In the context of the ideological battles of the Cold War, freedom of religion was a symbolic reason why South Korea was superior to North Korea, an ideological cause that the ROK must win the system competition with the 'northern puppet regime'. The pliable secular-religious distinction provided fertile ground, especially for Protestants, Catholics, and Buddhists, not only to vitalize the religious field whose liberty should be protected, but also to renew the public impact of their respective organizations. Among the three religious traditions, Protestantism and Catholicism had the closest relations to the US military authorities and the Syngman Rhee government, both of which, in turn, offered the two Christian traditions a great deal of institutional and material support (Kim 2013, pp. 314–17).

The construction of porous secularity has brought about a far-reaching change in the social location of religion, inter-religious relations, and the contour of religious terrain. In contemporary South Korea coexist a variety of transcendent traditions and groups, ranging

from folk beliefs, e.g., fortune-telling and Shamanic traditions, to new indigenous religions that have appeared since the late 19th century, e.g., Ch'ŏndogyo, Taejonggyo, Won Buddhism, and the Unification Church, and to the so-called 'world religions', e.g., Buddhism, Protestantism, Catholicism, and Confucianism. A postcolonial shift to mild secularism not hostile to religions and the approving conceptualization of religion went hand in hand with the remarkable growth in religious population and organizations in the last half of the 20th century, which, after all, contributed to the formulation of a unique religious landscape in South Korea. One may notice that there are three striking features of the religious landscape in this East Asian society: first, approximately half of the entire population consists of people with religious membership, and about the other half of those with no religious affiliation; second, among religions, Buddhism, Protestantism, and Catholicism account for most (approximately 98%) of the entire religious population; and third, the overall religious population is by and large divided into the 'Eastern tradition', i.e., the Buddhist half, and the 'Western tradition', i.e., the Christian half (Baker 2006, pp. 249–75). The chart below displays the changes in religious affiliation for the twenty years between 1985 and 2005 in South Korea (Cho 2014, p. 324; T'onggyech'ŏng 2016).

|  | 1985 | | 1995 | | 2005 | |
|---|---|---|---|---|---|---|
|  | **Population** | **Rate** | **Population** | **Rate** | **Population** | **Rate** |
| Total Population | 40,419,652 | 100.00% | 44,553,710 | 100.00% | 47,041,434 | 100.00% |
| Religious Population | 17,203,296 | 42.56% | 22,597,824 | 50.72% | 24,970,766 | 53.08% |
| Buddhism | 8,059,624 | 19.94% | 10,321,012 | 23.27% | 10,726,463 | 22.80% |
| Protestantism | 6,489,282 | 16.05% | 8,760,336 | 19.66% | 8,616,438 | 18.32% |
| Catholicism | 1,865,397 | 4.62% | 2,950,730 | 6.62% | 5,146,147 | 10.93% |
| Confucianism | 483,366 | 1.20% | 210,927 | 0.47% | 104,575 | 0.22% |
| Other Religions | 305,267 | 0.75% | 354,819 | 0.79% | 379,143 | 0.8% |
| No Religious Affiliation | 23,216,356 | 57.44% | 21,953,315 | 49.27% | 21,865,160 | 46.48% |

The mainstream circles of Korean Protestantism and Catholicism had a high affinity with national leaders and high-ranking bureaucrats such as the president, vice-president, and prime minister, receiving various financial and material supports from their sister churches and partners in the Western developed countries as well as internalizing the anticommunist agenda within their theologies and institutions. Korean Protestants, particularly those from Pyongyang, the current DPRK capital, who had defected to the South due to the establishment of a socialist regime in the North after the liberation of 1945, played a pivotal role in making Protestantism an unwavering anticommunist base in South Korea (Kang 2007, pp. 513–67). Throughout the industrialization in the 1970s and 80s, the revival services of Protestant churches, Pentecostal ones in particular, in which traditional religious aspirations were often embraced, and their tight-knit 'cell groups' effectively provided many South Koreans with alternative places of belonging from which they could gain new life orientation, confidence, and identity in the rapidly changing urban settings. It turned out that in the 1980s, Protestantism, in the end, accomplished massive quantitative growth. In 1945, less than one percent of the entire Korean people had Protestant membership, but until the 2000s, it became approximately one-fifth of the South Korean population (Kim 2006, pp. 309–11, 318–25).

Memories of the long and harsh persecution by the Chosŏn Dynasty in the 18th and 19th centuries influenced the political attitude of Korean Catholic Church to avoid conflict with the state. As did the Protestant right-wingers, Catholic Christians, particularly the bishops and the ecclesiastical superiors in the Korean Catholic hierarchy, generally supported the authoritarian governments and their anticommunist campaign after the Korean War. Being aware that the making of the modern state was no longer assertively

secularist and did not exclude public participation of religions, however, Korean Catholic Church actively engaged in socio-political campaigns whose issues included the urban poor, human rights, the ecological environment, inter-Korean relations, and inter-religious dialogue in the 1970s, 80s, and 90s (Choo 2010, pp. 37–67). Meanwhile, conservative Protestant evangelicals received social and political criticism due to many schisms and scandals and their aggressive proselytizing activities at home and abroad (Cho 2014, pp. 317–18), but Catholicism was able to build a different image as a 'civilized religion' that was decent, pro-democracy, and religiously inclusive. With such a positive perception of a desirable religion for the national society, the Korean Catholic Church achieved a great deal of growth in the last decades of the 20th century.

Buddhism, one of the major Korean religions, went through profound internal conflicts, a kind of decolonizing process, in which "pro-Japanese" married monks and "traditional" celibate monks clashed with one another after the liberation in the 1950s. At the same time, like the Japanese colonial government, the USAMGIK and the ROK governments continued to violate the freedom of religion, interfering with the autonomous management of Korean Buddhist orders, particularly with the disposition of their property, such as temples and land estates, until the 1980s. Meanwhile, fancying itself as "Korea's representative traditional religion" and "the repository of the nation's heritage culture", Buddhism steadily grew so that it became the most popular religion in late 20th-century South Korea.[12] In recent decades, especially since the democratization of the 1980s, South Korean Buddhists have increasingly reinforced the spiritual and participatory dimensions of their own tradition for both Buddhists and non-Buddhists, providing temple-stay programs, Sŏn/Zen-cum-psychotherapeutic courses, and international development programs for underdeveloped countries in the Global South.

### 3.2. Limited Freedom of Religion, the Shadow of the Pliable Secularity

South Korean modernity was in many ways shaped by the interplay of traditional cultural imprints, historical path dependence, and increased global interconnectedness, which featured a Confucian view of ethic, omnidirectional American influence, sweeping urbanization and industrialization, strong anticommunist ideology, military dictatorships, and cultural and institutional remnants of Japanese colonialism. Such entangled modernity created the secular-religious dualism that *vertically* divided religious traditions within the growing religious field. In other words, while Buddhism, Protestantism, and Catholicism grew consistently in the 1960s, 70s, and 80s, Confucianism, Shamanic tradition/folk beliefs, and new indigenous religions, which fell short of fully meeting the criteria for being the so-called 'world religion', were mainly referred to as 'philosophy'/'civilization', 'superstition', or 'pseudo-religions' rather than recognized as 'full religion'. Unlike the three major religions that dominate the South Korean religious market, the secular-religious distinction has not been so flexible or favorable to many minority religions, including Confucianism, Ch'ŏndogyo, Taejonggyo, Shamanism, and newer religious movements. Eventually, their membership dwindled to about 2% of the total religious population of contemporary South Korea.

Confucianism was the state religion of the Chosŏn Dynasty; however, in South Korea today, Confucianism as a religious entity does not have a great cohesive force. As its key leaders devoted themselves to participating in politics and even antigovernment activities after the 1945 liberation, Confucianism came into conflict with the new ruling groups of the Republic. While Confucianism was no longer a religion directly connected to political power, the American military government and Syngman Rhee government maintained state regulations, which the Japanese colonizers had created, to subordinate Confucian properties and institutions such as *Hyanggyo* (鄕校, local Confucian schools) to the state institutions. In the meantime, fatal factional disputes within the Confucian elite circles went on for decades, which eventually made it vulnerable to state interference as well as caused the loss of members and socio-political influence (Kang 2013, pp. 464–527). On the other hand, the identity of Confucian tradition has been an unsettled issue. In con-

temporary South Korea, except for a few attempts by Confucian leaders to religionize it, both Confucian adherents and South Korean people at large mostly put Confucianism into the categories of 'heritage', 'philosophy', 'education', 'spiritual culture' rather than that of 'religion'. After all, during the Cold War, the influence of Confucianism as a religious organization declined significantly.

Ch'ŏndogyo and Taejonggyo were the popular new religions that did not fit well into the official category of religion during the Japanese colonial era. The two aboriginal religions came to enjoy freedom of religion in the liberated space of Korea; however, the number of their adherents declined in the unleashed religious field of the postcolonial society. The emancipation from imperial Japan would first come as an incredible opportunity for the two new religions, so they immediately made a great effort to revive themselves right after the liberation, actively participating in nation-building activities and the administration of the new ROK government. However, their nationalist identity encompassing both Koreas was not suitable for the intense ideological milieu of South Korea during the Cold War that saw North Korea as the 'main enemy' rather than as a 'brother state' on the Korean Peninsula. In these political conditions of the new independent country, Ch'ŏndogyo soon became stigmatized as a "left-wing commie" and could not have full political citizenship until the early 1960s (Kang 2013, pp. 393–99). Similar to the situation of Ch'ŏndogyo, Taejonggyo also suffered a reduction in the number of its members in this political fix. During the Korean War, the principal leaders of Taejonggyo were abducted to North Korea, and the community of Taejonggyo believers was, in turn, significantly dismantled by the devastating effects of the civil war. Meanwhile, the excessive political participation of Taejonggyo leaders came into direct conflict with Syngman Rhee, the president of the First Republic period (1948–1960). In the end, the two nationalist new religions significantly contracted after a series of internal strifes and the failure of organizational reforms in the 1950s, and have never restored their reduced membership thereafter (Kang 2013, pp. 456–62).

With the liberation of 1945, all the colonial laws and rules that repressively classified marginal religious groups or traditions as *Yusa chonggyo* (類似宗敎, pseudo-religion) or *Sagyo* (邪敎, evil religion) disappeared. In many respects, however, the religious field of the postcolonial society came to revolve around the so-called 'world religions', i.e., the three religions of Buddhism, Protestantism, and Catholicism in particular. A good number of South Korean 'modernists', who considered the Western developed societies as the benchmark of modernizing their fatherland, especially mainstream Protestant Christians, some of whom were politicians or high-ranking officers, continued to call folk beliefs and shamanic rituals *Mishin* (迷信, superstition) and newer millenarian movements originated from the Protestant tradition *Idan* (異端, heresy). The South Korean government, the Park Chung-hee regime in particular, even conducted *Mishin t'ap'a undong* (未信打破運動, anti-superstition campaign) as part of the New Community Movement against folk beliefs or Shamanic tradition until the late 1970s. Emergent religious movements such as the Unification Church founded in 1954 by MOON Sun Myung (文鮮明, 1920–2012) and the Olive Tree Church founded in 1955 by PARK Tae Son (朴泰善, 1917–1990), both of which appeared in the wake of the Korean War, were often called 'heresy' or 'evil religion' rather than 'religion'. In the case of the Unification Church, in the end, it had to seek survival and new avenues abroad, including in Japan and the USA.

In short, the flexible secularism maintained by the authoritarian state was a critical contributor to the positive conceptualization of religion as well as the reformulation of the religious sector in Cold War South Korea. While the postcolonial rise of permeable secularity provided a favorable environment for an increase in the national religious membership, it did not result in the formation of an anarchic religious market. It was, rather, selectively applied to religious groups, thereby rearranging the religious terrain hierarchically. Buddhism, Protestantism, and Catholicism enjoyed the affirming semantics of religion, becoming the major religious groups that have now occupied most of the religious population in contemporary South Korea. It is paradoxical, however, that the positive conception of religion and the flexible form of secularity went with the strong suspicion and negation

of the identity and role of marginal religious traditions in the context of the East Asian urbanization and industrialization, folk or Shamanic traditions, and new religious groups in particular. The social space for these minority religions was greatly curtailed in the latter half of the 20th century. Since the 1990s, when democratic governments began to take office, there has been a gradual increase in public and governmental efforts to recognize the religious liberty of minority religious groups as well as to promote cultural values and features of marginal folk or Shamanic traditions.

## 4. Conclusions

In this article, through the South Korean case of secularity, I showed that, first, the way to distinguish between religion and secular is not fixed but varies depending on cultural, historical, and glocal context; second, in modern society, religion and secular use and 'help' each other to constitute themselves; and third, in the Cold War geopolitics, the politics of secularism was deeply related to the ideological struggle of postcolonial nation-building.

The rigid colonial secularism that had banned the public and political engagement of religion was replaced by the flexible secular-religious divide in liberated South Korea. Through the processes of industrialization, urbanization, and democratization, social fields and institutions have been significantly differentiated according to their secular functions in the Republic of Korea. Unlike in postwar China and Japan, a good portion of Korean modernist intellectuals and major nationalist leaders viewed religion as an important requisite, a significant modern program that could help save and advance the national community in crisis.

The porous mode of secularity admitted religious groups to participate in processes of national development, especially in the aftermath of the Korean War. In the postcolonial society, religions were largely invited to play crucial public roles between the state and the economic sphere in combating enemies within and outside, including feudalistic vestiges of 'pre-modernity', poverty, illiteracy, military dictatorship, and atheist North Korean communists. The role of religion was not limited to the organizational level. Behind the secular and materialist achievement, Confucian ethic and familism have been applied in motivating, pushing, and/or justifying South Koreans to devote themselves to developing their villages and companies, as well as transforming the national community as a whole. The authoritarian developmental state, conglomerates such as Samsung and Hyundai, and various formal and informal organizations all actively used Confucian values and aspirations, such as the image of family, loyalty, virtue, and self-discipline, in worldly economic development, social and political engineering, organizational management, and even marketing.

With the liberation, State Shintō immediately disappeared from the Korean Peninsula. The postcolonial turn of disenchantment seemed to provide transcendent traditions and emergent redemptive movements with a newly liberated religious space in which to revitalize themselves. In the context of Cold War ideological struggle, having a religious affiliation, especially a Christian one, was often crucial for securing one's citizenship during the nation's critical junctures, such as the Korean War and the military dictatorship in pre-democratization South Korea. It is certain that the pliable form of secularity, combined with the ideological and institutional conditions of the Cold War, led to significant organizational changes in the religious field.

Such a pro-religious social environment, underpinned by the codification of freedom of religion, contributed to the reformulation of the nation's religious diversity in a twisted way. On the one hand, Buddhist and Christian populations grew remarkably in the liberated religious field, with religious liberty widely recognized as a key principle of modern civilization. Many South Koreans who were 'wounded' in the process of rapid industrialization found comfort in Protestant churches—especially Pentecostal or charismatic ones—and learned religious techniques to empower themselves in the newly urbanized environment. Meanwhile, Buddhism was firmly established as a 'treasure trove' of the nation's cultural heritage and thereby as a 'representative national religion'.

Amid modernization, strong anticommunist ideology, and public religious discourses centered on the mainstream religions, on the other hand, folk beliefs, nationalistic new religions, and newer religious movements—especially those driving from Protestant Christian tradition after the liberation or the Korean War—were often considered "superstitions", "pseudo religions", "heretics", or "evil religions". As a result, they were completely marginalized in the field of religion. In short, the porous secularity was two-faced and not so flexible or accommodating to these religious minorities. While freedom of religion was applied selectively, the religious sphere was reformulated *vertically* again in the postcolonial society.

**Funding:** This article results in part from research conducted at the Kolleg-Forschergruppe (KFG) Multiple Secularities—Beyond the West, Beyond Modernities, at Leipzig University. The KFG is funded by the German Research Foundation (DFG). This work was also supported by the Ministry of Education of the Republic of Korea and the National Research Foundation of Korea (NRF-2023S1A5A2A03086236).

**Institutional Review Board Statement:** Not applicable.

**Informed Consent Statement:** Not applicable.

**Data Availability Statement:** Data is contained within the article.

**Acknowledgments:** I wish to thank Monica Wohlrab-Sahr and Judith Zimmermann for supporting this research at Leipzig University. I am also grateful to Yijiang Zhong, Christoph Kleine, James Spickard, Johannes Duschka, and other anonymous reviewers for their helpful comments on an earlier version of the article.

**Conflicts of Interest:** The author declares no conflict of interest.

## Notes

[1] Some of the scholary literatures of rethinking secularization include the following: (Asad 2003; Berman et al. 2013; Calhoun et al. 2011; Dreßler 2019; Mack et al. 2009; Taylor 2007; Warner et al. 2010; Wohlrab-Sahr and Burchardt 2012).

[2] Chinese, Japanese, and Korean peoples commonly use '宗教' to translate 'religion', but pronounce it differently from each other as *Zongjiao, Shukyō,* and *Chonggyo*.

[3] Arguing that ideological struggles shape state policies toward religion, Ahmet T. Kuru suggests two types of secularism, i.e., passive secularism and assertive secularism. "Assertive secularism" requires the state to play an assertive role to exclude religion from the public place and to impose secularism as a comprehensive doctrine at the expense of religion. "Passive secularism" demands the state to play a passive or neutral role by accepting the public visibility of religion. According to Kuru, France, Mexico, and pre-Erdoğan Turkey are the countries where supporters of assertive secularism is dominant, while the United State, India, and the Netheranes are cases where passive secularism is dominent. (Kuru 2009, pp. 6–14).

[4] Since then, the American troops have been stationed across South Korea until now.

[5] The National Charter of Education (國民教育憲章) of the ROK, which was proclaimed in 1968 and abolished in 1994, begins with this sentence "We have been born into this land charged with the historic mission of regenerating the nation" and ends with "Looking forward to the future when we shall have the honorable fatherland unified for the everlasting good of posterity, we, as an industrious people with confidence and pride, pledge ourselves to make new history with untiring effort and collective wisdom of the whole nation". The National Charter of Education was criticized for being the South Korean version of the Imperial Rescript on Education (教育ニ関スル勅語) that had been signed by Emperor Meiji of Japan in 1890.

[6] Dong Choon Kim categorizes the mass killings from June 1950 to July 1953 into three types—the first type includes thoses cases committed by military forces in the course of military operations, the examples of the second fomr are the ROK government's executions of "suspicious civilians" or political prisoners, and the third mode is comprised of political or personal reprisals by civilians and yourth groups being sponsored by the ROK govenment (Kim 2004, pp. 529–40).

[7] The urbanization rate of South Korea soared from 28% in 1960 to 74.4% in 1990. The number of cities as the administrative districs increased from 27 in the 1960s to 73 in the 1990s (Hong et al. 1996, pp. 81–82).

[8] Ibid. pp. 81–82.

[9] From 1962 to 1996, Park Chung-hee, a symbol of South Korean modernization, and the two other former ROK presidents, pushed for the five-year economic development plan (經濟社會發展5個年計劃) whose model was the planned economy projects of *Manchukuo*. See (Han 2005, pp. 163–83).

[10] *Manchukuo* was the place that conceived both leaderships of North and South Korea. The core power elites of the DPRK have been the former guerilla and their descendants that survived the chase of the *Manchukuo* regime. After graduation from the Japanese military academy, Park Chung-hee began his career as an officer of the Japanese Manchurian army before the liberation of 1945. See (Duara 2004, pp. 245–54).

[11] "Another family" (또 하나의 가족) was the key slogan of Samsung group from 1997 to 2007.

[12] The chart below displays the change of religious affiliation for the twenty-year period between 1985 and 2005 in South Korea (Cho 2014, p. 324).

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
