# Peer review of "Porous Secularity: Religious Modernity and the Vertical Religious Diversity in Cold War South Korea"

_religions, doi:10.3390/rel15080893_

Round 1

Reviewer 1 Report

Comments and Suggestions for Authors

This article offers a valuable analysis of the complex interplay between religion and secularism in South Korea's modernization process. The article challenges the simplistic notion of secularization as a decline of religion. However, the clarity and focus could be improved to strengthen the argument.

Clarity: The text would benefit from a more accessible introduction to secularization and its limitations. Dense sentences with multiple keywords could be unpacked into clearer paragraphs with well-defined topic sentences and supporting examples. For instance, lines 141-142 regarding the decline of community sentiment could be elaborated on with specific historical events or data.

Focus: While the South Korea case study is strong, the broader question of globalization’s impact on religion and secularism in non-Western societies needs a clearer connection.

Examples: Including concrete historical examples of how religious groups in South Korea navigated the concept of secularism during the Cold War would enhance the argument.

Main issue: The author puts too much content and various themes into one paragraph. Virtually every paragraph has 3-4 sentences that could be unpacked. Choose only 1 of these sentences and write a paragraph about it. 

To provide just a few examples:

lines 141-142: “a great degree of community sentiment ….disappeared.” => how so? Why? You could write an entire paragraph explaining this statement.

And lines 149-152: what data are these statements based on? Add at least a footnote.

Line 161: why is Kendall cited here? state the reason.

Line 163-165: Explain what you mean in a full paragraph. Don’t just make a statement, provide examples to support your statement. 

Line 177: Confucianism was not only politically used post-war; Japanese colonizers used it as an instrument too. (see also line 216)

Line 290: that’s the beginning of a new paragraph

Line 338: which survey or census data is this statement based on? Provide footnote.

Line 362: please define Confucianism. Is it a “state religion”? a “state doctrine”? or something else? A combination of various factors?

Lines 449-452: this sentence alone could be the topic for an entire paper!

Line 479: the religious sphere was organized "vertically" – what does this mean?

Comments on the Quality of English Language

English is ok.

Author Response

1. Clarity: The text would benefit from a more accessible introduction to secularization and its limitations. Dense sentences with multiple keywords could be unpacked into clearer paragraphs with well-defined topic sentences and supporting examples.

Response: The focus of the introduction is to introduce readers to the purpose of the paper showing the way in which ‘religion’ and ‘secular’ are distinguished from each other in South Korea, and at the same time have needed one another in order to distinctively construct themselves. I revised the introduction for the readers accordingly.

2. For instance, lines 141-142 regarding the decline of community sentiment could be elaborated on with specific historical events or data.

Response: I revised that sentence and put a footnote supporting it.

3. Focus: While the South Korea case study is strong, the broader question of globalization’s impact on religion and secularism in non-Western societies needs a clearer connection.

South Korea was used as a case showing how local transformation of the secular-religious distinction was shaped by global forces and their interactions within and beyond East Asia. In this article, I showed how the idea of secular and the category of religion, both of which had been introduced to Korea from outside in the 19th century, were changed in South Korea as East Asia entered into the newly globalized Cold War order after World War II.

4. Examples: Including concrete historical examples of how religious groups in South Korea navigated the concept of secularism during the Cold War would enhance the argument.

Response: This paper focuses on describing how the transformation of secularity, that is, the secular-religious differentiation, was achieved and how the religious changes in postcolonial South Korea are connected to these changes. This paper does not examine religious groups' navigation of the concept of secularism.

5. Main issue: The author puts too much content and various themes into one paragraph. Virtually every paragraph has 3-4 sentences that could be unpacked. Choose only 1 of these sentences and write a paragraph about it.

Response: This article attempts to identify the general form of secularity constructed in South Korea after 1945 liberation and its religious consequences and implications. In this paper I focus on distinguishing the “Korean pattern” of secularity, which I argue is porous or flexible. So this paper is, inevitably, an overview of that pattern. I took several examples to show the context in which such a pattern was formed, and in some cases, I had to put several examples in one paragraph.

To provide just a few examples:

6. lines 141-142: “a great degree of community sentiment ….disappeared.” => how so? Why? You could write an entire paragraph explaining this statement.

Response: I revised that sentence and put a footnote supporting it.

7. And lines 149-152: what data are these statements based on? Add at least a footnote.

Response: I put a footnote with data supporting it.

8. Line 161: why is Kendall cited here? state the reason.

Response: The quotation from Kendall’s book shows that while the state suppressed shamanic rituals performed in urban outskirts and rural areas as ‘superstitions’, i.e., the enemy of national development, industrialization evaporated the shamanistic sacredness that Koreans had felt daily from the natural environment, especially its overwhelming energy.

9. Line 163-165: Explain what you mean in a full paragraph. Don’t just make a statement, provide examples to support your statement.

Response: This sentence aims to summarize the contents of 136-161. Also, Gangnam was cited as an example of the highly urbanized culture in contemporary South Korea.

10. Line 177: Confucianism was not only politically used post-war; Japanese colonizers used it as an instrument too. (see also line 216)

Response: I explore the pattern of religious-secular differentiation formed in post-liberation South Korea. In this paper, Confucianism is cited as a factor that contributed to the formation of that pattern of secularity.

11. Line 290: that’s the beginning of a new paragraph

Response: The three striking features that Dornbaker suggests are presented as the characteristics of South Korea's unique religious landscape, which SK’s porous secularity contributed to its formation.

12. Line 338: which survey or census data is this statement based on? Provide footnote.

Response: Census data is now provided as a footnote.

13. Line 362: please define Confucianism. Is it a “state religion”? a “state doctrine”? or something else? A combination of various factors?

Response: Since being introduced from China, Confucianism has been a major tradition in Korea, which has changed its form depending on the times. Koreans during the Joseon Dynasty appropriated it as a ‘state religion.’ After the liberation, it functions as a ‘cultural heritage’ or an influential ethical system in South Korea.

14. Lines 449-452: this sentence alone could be the topic for an entire paper!

Response: This paper aims to show how the distinction between ‘secular’ and ‘religion’ in South Korea is flexible, passive, or porous at both the organizational and semantic levels. For this purpose, I use Confucian values as the religious elements that at a semantic level influence the formation of secularity in post-colonial South Korea.

15. Line 479: the religious sphere was organized "vertically" – what does this mean?

Response: After liberation, freedom of religion became a constitutional provision, but the religious sphere was not a space where various religious traditions coexisted equally or anarchically. In Cold War South Korea, the freedom of major religions was more respected and supported, while the freedom of minority religions was often ignored and infringed. While Christinity and Buddhism were provided with more state support, folk/Shamanic faith and new religions had their religious freedom violated by the state. Ultimately, this resulted in a vertical, unequal, and hierarchical order among religions.

Reviewer 2 Report

Comments and Suggestions for Authors

The author of the paper titled “Porous Secularity: Religious Modernity and the Vertical Religious Diversity in Cold War South Korea” discusses religious diversity in the secular context of South Korea and writes from a Confucian perspective of the role of Confucianism in the economic development of the nation. In its current form the article remains vulnerable to scholarly challenges, and if the article is to be published, it will need to include a scholarly discussion on the following:

1) The author argues that one of the major reasons for the economic development of South Korea is the role of Confucian values aiding the push for modernisation (lines 167-197). However, the foundation for this viewpoint ought to be thoroughly supported with scholarly evidence, as there are currently numerous competing viewpoints on the matter. The author should also elaborate on the intertwined relationship between the culture of Confucianism and the religion of Confucianism in South Korea. There are many elements in Buddhism, Shamanism and Confucianism which can be seen as culture of South Korea.

2) Christianity has been present in Korea for over 150 years, and the role of Christianity in Korean history has been significant, culturally, politically and economically. The author ought to give a fuller account of the contribution it has made to Korea from a scholarly perspective (lines 323-329).

3) The author ought to be clear and disclose the relationship between conglomerates and the state at the time of military dictatorships in South Korea, and of the rampant corruption and abuse of power that existed in society, which has been well-established by scholars (lines 433-452). If this is not clearly explained, there is a danger that Confucianism could be seen to be blind to or even contribute to such issues from the lack of critical analysis.

4) Overall, taking an overwhelming Confucian viewpoint to write on religious diversity has ended up with the author not providing a fuller perspective of the issue. The article over-generalises and glosses over the impact of Christianity in many places and numerous arguments are not thoroughly supported with scholarly evidence. Should the author desire to retain a Confucian-centric view, the title ought to be changed to reflect its content and the scope of the article be narrowed accordingly.

Author Response

1. The author argues that one of the major reasons for the economic development of South Korea is the role of Confucian values aiding the push for modernisation (lines 167-197). However, the foundation for this viewpoint ought to be thoroughly supported with scholarly evidence, as there are currently numerous competing viewpoints on the matter. The author should also elaborate on the intertwined relationship between the culture of Confucianism and the religion of Confucianism in South Korea. There are many elements in Buddhism, Shamanism and Confucianism which can be seen as culture of South Korea.

Response: The goal of this paper is to look at the re-formation of secularity or the reformulation of religious-secular divide in South Korea after liberation, and to look at the resulting reorganization of the religious sphere in general. Through this, this paper seeks to contribute to the literature related to secularities. According to Charles Taylor, modern society is characterized by a transition from religious norms to an ethical order led by secular values. The process of declining religious influence is often called secularization. This paper presents this formation of Western secularity as one of the different forms of ‘secular-religious divide’ on earth.

In the process of South Korea’s modernization, the influence of Confucianism has generally declined. However, many worldly or secular choices were made according to Confucian ethics, whether the results were negative or positive. South Korea's modernization after liberation was also a process of de-Confucianization. I don't deny this part. I agree that the secular/functional transformation of South Korean society is partly motivated by Confucian personal or communal ethics. Religion made both functional or positive and reactionary or negative contributions to the formation of South Korea's modernity, thereby making South Korea's secularity complex and paradoxical in the Cold War era

2. Christianity has been present in Korea for over 150 years, and the role of Christianity in Korean history has been significant, culturally, politically and economically. The author ought to give a fuller account of the contribution it has made to Korea from a scholarly perspective (lines 323-329).

Response: This paper does not focus on the specific role of Christianity. The goal of this paper is to understand the re-formation of Korean secularity, that is, the postcolonial pattern of  secular-religious differentiation, and to consider the resulting reorganization of South Korea's religious landscape. In particular, I am interested in examining the  cultural/historical/political/transnational conditions that have shaped Korea's current secularity.

I agree that while South Korean society was becoming secularized, or functionally differentiated, the religious population, including Christians, actually increased, and after the Korean War, Christianity played a great role in public areas such as education, medicine, media, and social welfare. However, the goal of this paper is not to repeat or elaborate on this topic, which has already been extensively studied. This paper focuses on describing the pattern of ‘flexible’ or ‘religion-friendly’ secularity formed in South Korea. The goal is to make people realize that South Korea's position in global society and Korea's cultural and historical conditions contributed to the formation of such a unique secularity.

3. The author ought to be clear and disclose the relationship between conglomerates and the state at the time of military dictatorships in South Korea, and of the rampant corruption and abuse of power that existed in society, which has been well-established by scholars (lines 433-452). If this is not clearly explained, there is a danger that Confucianism could be seen to be blind to or even contribute to such issues from the lack of critical analysis.

Response: The author does not deny the ‘authoritarianism,’ ‘corruption,’ and ‘non-democracy’ that accompanied South Korea’s modernization. And, I recognize that Confucianism had both positive and negative roles in the modern transformation of South Korea. The reason why the author exposed the authoritarianism, undemocratic nature, and corruption of South Korean politics and economy was to show how the ‘postcolonial phenomenon’ is institutionally and culturally linked to the path dependence of Japanese colonialism.

4. Overall, taking an overwhelming Confucian viewpoint to write on religious diversity has ended up with the author not providing a fuller perspective of the issue. The article over-generalises and glosses over the impact of Christianity in many places and numerous arguments are not thoroughly supported with scholarly evidence. Should the author desire to retain a Confucian-centric view, the title ought to be changed to reflect its content and the scope of the article be narrowed accordingly.

Response: The goal of this article is to discern the formation of ‘secular-religious differentiation’ or ‘secularity’ in postcolonial South Korea and the changes in the religious landscape that this has resulted in. The author does not claim that Confucianism made a significant contribution to the formation of religious diversity in South Korea. It is not the goal of this article to point out again the significant influence of Christianity in various areas of South Korean society, such as schools, hospitals, and politics.

I tried to help readers see the context in which the ‘secular’ and ‘religious’ spheres in South Korea are distinguished in a somewhat mutually open and dependent or ‘mutually porous’ way rather than being mutually hostile. In other words, the differentiation between religion and the secular in South Korea operates as a condition for interaction between the two. Again, religion and secular in South Korea are structurally and semantically connected. Although they ‘hostile’ each other, they refer to and utilize the other in order to construct themselves. The interaction between South Korea's historical, cultural, and global conditions, including colonial experience, decolonization, the Cold War, and religious traditions, contributed to the emergence of such flexible secularity.

The emergence of ‘porous secularity’ after liberation contributed to the formation of a vertical or discriminatory religious order rather than an equal or anarchic one. Ultimately, religious diversity centered on Christianity and Buddhism, which was completely different from the religious landscape that had appeared during the Japanese colonial era, was formed in South Korea after liberation.

Round 2

Reviewer 1 Report

Comments and Suggestions for Authors

looks perfect, great!

Reviewer 2 Report

Comments and Suggestions for Authors

The author has addressed the issues in the revised paper and I recommend it for publication.